# Current and Emerging Radiotracers and Technologies for Detection of Advanced Differentiated Thyroid Cancer: A Narrative Review

**DOI:** 10.3390/cancers17030425

**Published:** 2025-01-27

**Authors:** Reza Pishdad, Prasanna Santhanam

**Affiliations:** 1Division of Endocrinology, Diabetes & Metabolism, Massachusetts General Hospital, Harvard Medical School, Boston, MA 02114, USA; 2Division of Endocrinology, Diabetes & Metabolism, The Johns Hopkins University School of Medicine, Baltimore, MD 21224, USA; psantha1@jhmi.edu

**Keywords:** PSMA-based imaging, RGD-based imaging, radioiodine avid DTC, radioiodine-refractory DTC, theranostics, iodine-based radiotracers, [18F] FSO3, Ga-68 based SSTR imaging, 68Ga-DOTA-FAPI-04 PET/CT

## Abstract

Differentiated thyroid cancer is a common type of thyroid cancer, but its diagnosis and treatment become challenging in advanced stages, especially when traditional imaging methods are less effective. This review explores innovative imaging agents called radiotracers, which can improve the detection of DTC and provide more precise information about the cancer’s behavior. By highlighting the strengths and future potential of these advanced imaging techniques, the authors aim to help researchers and healthcare providers better diagnose and manage thyroid cancer. These advancements could pave the way for personalized treatments and improved outcomes for patients with advanced DTC, while also inspiring further research in this area.

## 1. Introduction

Differentiated thyroid cancer (DTC), which includes papillary and follicular thyroid cancers, differs significantly in pathology compared to other thyroid malignancies such as medullary thyroid cancer (MTC), anaplastic thyroid cancer (ATC), and Hurthle cell carcinoma. DTC arises from thyroid follicular cells and is characterized by its well-differentiated cellular architecture and a generally indolent course. It typically retains the ability to uptake iodine due to the presence of sodium–iodide symporters, making radioiodine therapy a viable treatment. In contrast, MTC originates from parafollicular C-cells and is associated with calcitonin secretion, lacking the iodine uptake pathway. ATC represents an undifferentiated, highly aggressive form of thyroid cancer with rapid growth, resistance to most therapies, and extremely poor prognosis. Hurthle cell carcinoma, a variant of follicular thyroid cancer, is noted for its abundant mitochondria and eosinophilic cytoplasm, often displaying a more aggressive clinical course than conventional DTC [1]. These distinct pathological features among thyroid cancer subtypes underpin differences in clinical presentation, diagnostic approaches, and treatment strategies.

The diagnosis and management of advanced DTC present ongoing challenges due to the limited sensitivity of traditional imaging modalities. Recent advancements in radiotracer technology have provided alternatives for detecting and characterizing metastatic lesions in patients with DTC. Radiopharmaceuticals play critical roles in both diagnosing and treating advanced DTC, with certain agents serving primarily diagnostic purposes and others being therapeutic. Diagnostic radiopharmaceuticals include I-123, I-124 PET/CT, and novel tracers like [18F] TFB, [18F] FSO3, and 68Ga-based somatostatin receptor (SSTR) imaging agents, which are primarily utilized to visualize disease burden, assess metastases, and guide therapeutic decisions. On the other hand, therapeutic radiopharmaceuticals, such as I-131 and emerging agents like 177Lu-DOTA-RGD2 and 68Ga-DOTA-FAPI-04, are designed to deliver targeted radiation to cancer cells, capitalizing on molecular and cellular pathways unique to thyroid cancer [2,3]. This review explores the new radiotracers, their mechanisms of action, and their clinical utility in the context of advanced DTC. The aim is to provide insights into the strengths and limitations of these imaging agents to improve diagnostic accuracy and therapeutic decision-making for patients with advanced DTC.

## 2. Radioiodine-Avid DTC

### 2.1. NIS (Sodium-Iodide Symporter)-Based Imaging: Iodine-Based Radiotracers (I-123, I-131, I-124 PET/CT)

NIS mediates iodide uptake in thyroid cells, which is essential for the effectiveness of radioactive iodine-based imaging and therapy in treating DTC [4,5,6,7]. This therapy is effective in DTCs that express NIS, allowing for targeted destruction of cancerous cells [4,5,6,7]. NIS expression is regulated by thyroid-stimulating hormone (TSH), which enhances iodide uptake in thyroid cancer cells. The regulation of NIS expression and membrane insertion is carried out by different pathways in different tissues (for example, Retinoic acid upregulates NIS expression in breast tissue) [5,6]. Additionally, NIS is expressed in several non-thyroidal tissues, including the salivary glands, stomach, and the breast, as outlined above, leading to the physiological distribution of iodide-based tracers and, consequently, imaging findings beyond the thyroid tumor cells [7,8,9]. Despite this promise, many DTCs exhibit reduced or heterogeneous NIS expression, leading to decreased radioiodine therapy effectiveness [5,6,10,11]. Even in cases where NIS mRNA levels are high, the protein may not be correctly localized to the cell membrane, diminishing iodide uptake. Mutations associated with RAI-R (Radioiodine-Refractory) status include BRAF V600E, TERT (telomerase reverse transcriptase gene) promoter, and TP53 (tumor protein) mutations, while RET (rearranged during transfection) fusions and RAS (Rat Sarcoma) mutations are more frequent in the RAI-A (Radioiodine Avid) disease [12]. RTKs, RAS, BRAF, and TERT genetic aberrations lead to structural protein changes that activate the mitogen-activated protein kinase (MAPK) and phosphatidylinositol-3-hydroxykinase (PI3K) signaling pathways, leading to dedifferentiation and NIS dysfunction leading thyroid cells to lose RAI avidity [13].

Iodine-based radiotracers, including I-123, I-131, and I-124 PET/CT, play a crucial role in the diagnosis and management of radioiodine-avid DTC by binding to the NIS. Iodine-123, a gamma emitter, is employed for diagnostic imaging due to its favorable pharmacokinetics (Table 1) [14]. In 14 patients with DTC involving 35 foci, 50 MBq (1.5 mCi) of I-123 detected more lesions when compared with 111 MBq (3 mCi) I-131 for whole-body scanning in patients [14]. Research indicates that I-123 doses between 111 and 185 MBq achieve high sensitivity for imaging, detecting 86.7% of cases (26 out of 30) when compared to early post-I-131 treatment scans. This dosage also enables extended 48 h imaging without the risk of “stunning”, preserving diagnostic accuracy [15]. However, diagnostic imaging using relatively low doses of I-123 may not consistently predict subsequent therapeutic uptake of I-131, especially for metastases in lymph nodes and lungs in cases of DTC [16]. Pre-therapy scans utilizing I-123 have been shown to be more effective in detecting locoregional metastases or remnants compared to post-therapy I-131 imaging obtained after seven days, likely due to the extended delay before obtaining post-therapy imaging (Of more than 140 lesions, 35% were detected only by I-123 pre-therapy images compared with 15% detected only on post-therapy images) [17]. Pre-therapy scans provide valuable insights into unsuspected lymph nodes or distant metastases, allowing for the adjustment of treatment dosing accordingly [18]. A review of 122 patients at Yale New Haven Hospital found that pre-treatment scans provided valuable information in 25% of cases [18]. However, pre-treatment scans may not affect the radioiodine I-131 dosage used for remnant ablation in DTC or influence recurrence rates, particularly in low-risk tumors, especially in community practice settings [19]. Our study found no significant difference in the mean radioactive iodine dose for remnant ablation between patients who underwent pre-therapy scans (97.56 ± 27.98 MBq) and those who did not (97.23 ± 32.40 MBq, *p* = 0.45), particularly in the community setting, where higher doses were commonly administered before guideline changes [19]. Additionally, false-positive imaging results can occur, as seen in some patients with PTC and intraosseous hemangiomas (IH) [20].

Iodine-131 (I-131) therapy and subsequent post-therapy scanning are the gold standard in I-131-based imaging. However, pathological false-positive I-131 scans can lead to diagnostic errors and inappropriate treatment. A retrospective study aimed to evaluate the optimal imaging modality for diagnosing pathological false-positive I-131 scans in a cohort of DTC patients [21]. Out of 156 patients, six cases of pathological false positives were identified, with an incidence of 3.85%, and subsequently, MRI demonstrated superior soft tissue resolution and was identified as the most effective imaging modality for diagnosing these pathological false positives [21].

### 2.2. Use of Single-Photon Emission Computed Tomography/Computed Tomography (SPECT/CT)

I-131 SPECT/CT scanning seems to show superior accuracy in detecting cervical lymphadenopathy and predicting relapses by the nodal metastasis (NM) stage compared to whole-body scans (WBS) in PTC. A prospective study designed by Malamitsi and colleagues compared the diagnostic performance of I-131 post-ablation whole-body scans (WBS) and SPECT/CT scans in 58 patients with PTC [22]. Both scans were performed the same day after total thyroidectomy and I-131 remnant ablation, with SPECT/CT results confirmed by ultrasound and, in specific cases, by CT and histology. Statistically significant improvements with SPECT/CT were observed in detecting cervical lymphadenopathy (*p* = 0.031) and in predicting relapse by the NM stage (*p* = 0.033) over a 5-year follow-up in 36 patients [22]. Utilizing semi-quantitative I-131 analysis with early and delayed whole-body scans offers supplementary insights into assessing ablation success, potentially facilitating the evaluation of treatment response in metastasis [23].

**Table 1 cancers-17-00425-t001:** Comparison between I-123 pre-treatment and I-131 post-treatment scan.

Aspect	I-123 Pre-Treatment Scan	I-131 Post-Treatment Scan
Disease Burden Estimation	Underestimates disease burden, especially in patients with prior radioiodine therapy or distant metastasis [16,24].	More accurate in assessing disease burden post-treatment.
Clinical Information	Provides crucial details on thyroid remnants and unsuspected metastases, aiding in I-131 dosing and planning [17,18].	Confirms treatment outcomes and disease status.
Concordance with Metastases Detection	High concordance with I-131 for thyroid bed and bone metastases; lower for lymph node and lung metastases [16,25,26].	More accurate in detecting all metastases post-treatment.
Image Quality	Better image quality and resolution, useful for locoregional metastases and thyroid remnants [14,15].	Lower image quality; primarily used to assess treatment efficacy.
Impact on Clinical Management	Can alter treatment plans, such as adjusting I-131 doses or using two-step ablation protocols [18,27].	Validates treatment effectiveness and informs follow-up decisions.

### 2.3. I-124 PET/CT Scan

I-124 PET/CT imaging leverages the unique properties of the isotope Iodine-124 (I-124) for enhanced thyroid imaging and cancer detection. I-124, a positron-emitting radioisotope with a half-life of about 4 days, enables longer imaging windows and higher resolution due to its favorable decay characteristics [28].

Studies comparing the diagnostic performance of various imaging modalities highlight the superiority of I-124 PET/CT over conventional methods like I-131 WBS, standalone PET, and CT scans. In an early study involving 12 patients, I-124 PET/CT was very effective in detecting lesions compared to I-131 WBS (100% vs. 83%, respectively) [29]. Accurate quantification in I-124 PET imaging requires meticulous determination of absolute recovery measurements tailored to the specific PET scanner and radionuclide employed for accurate dosimetry [30]. I-124 PET/CT demonstrates sensitivity in detecting radioiodine-avid lesions in DTC, even in cases where lesions are not visualized on post-treatment I-131 scans [24]. In a systematic review by our group involving 141 patients and 415 lesions, the pooled sensitivity/specificity was greater than 94% and 49%, respectively [24]. The potential of I-124 PET/CT in identifying lesions for I-131 therapy warrants further dosimetric studies to establish its clinical role comprehensively.

Retrospective dosimetry calculations using I-124 PET/CT imaging have revealed a correlation between higher lesion average dose (AD), biological effective dose (BED), and equivalent uniform dose (EUD) values and therapeutic response to I-131 therapy [31]. These findings indicate the potential of I-124 PET/CT in predicting lesion response and optimizing treatment strategies for DTC patients [31].

### 2.4. [18F] Fluorosulfate ([18F] FSO3) and [18F] Tetrafluoroborate ([18F] TFB)

The utilization of novel radiotracers such as [18F] tetrafluoroborate ([18F] TFB) and [18F] fluorosulfate ([18F] FSO3) represents an advancement in the molecular imaging of radioiodine-avid DTC [32]. Fluorosulfate exhibits a high affinity for the human NIS, facilitating the synthesis of the radiotracer [18F]-SO3F- [33]. This tracer can be synthesized with high yield and specific activity, making it suitable for both preclinical and clinical PET imaging of NIS expression and thyroid-related diseases. The images of normal mice show uptake in the thyroid, salivary glands, and stomach mucosa [33].

[18F] TFB, on the other hand, exhibits a radiation exposure level comparable to other 18F-labeled radiotracers and demonstrates a biodistribution pattern similar to 99mTc-pertechnetate, an established NIS tracer [34]. In this pilot study, [F]TFB-PET was shown to be comparable to I-124-PET for detecting thyroid cancer and its metastases (In a lesion-based analysis, both tracers performed equally well [41 vs. 40 foci], respectively) [35]. Additionally, it successfully identified viable DTC metastases that were not detected by [I]NaI-PET [35].

Ventura and colleagues evaluated the effectiveness of ([(18)F] TFB) PET for DTC in comparison to high-activity [(131)I] iodine scintigraphy, SPECT/CT, and [(18)F]FDG PET/CT [36]. A total of 26 patients receiving high-radioactive iodine therapy (range, 5.00–10.23 GBq) were included, with scans conducted after administering a median dose of 321 MBq of [(18)F] TFB. In total, 62 suspected lesions were identified: 30 were positive for [(131) I], 32 for [(18)F] TFB, and 52 for [(18)F] FDG. Among the iodine-positive lesions, three were false positives. A strong correlation was observed in tumor-to-background ratios at 40 and 90 min (Pearson correlation coefficient 0.91, *p* < 0.001) [36]. A negative correlation was found between [(18)F] TFB and [(18)F] FDG uptake (Pearson correlation coefficient −0.26, *p* = 0.041), which is not surprising since FDG PET CT avidity is associated with loss of NIS expression. In conclusion, [(18)F] TFB PET/CT may help predict DTC lesion positivity on iodine scans [36].

## 3. Radioiodine-Refractory DTC

### 3.1. 18F-FDG PET/CT

The use of 18F-FDG PET/CT is vital for the management of thyroid malignancies, particularly in cases of RAI-R DTC. The presence of the BRAF V600E mutation in PTC has been correlated with increased FDG PET avidity and higher standardized uptake value (SUV) compared to cases without this mutation [37]. In our systematic review and meta-analysis evaluating the effectiveness of FDG PET/CT in detecting residual disease in PTC patients with the BRAF V600E mutation, data from 12 studies published between 1995 and 2017 was analyzed. Of the 1144 patients in the pooled cohort, 843 had the BRAF V600E mutation, while 301 did not. Results showed that patients with the BRAF V600E mutation had a significantly higher likelihood of FDG-avid lesions, with a pooled OR of 2.12 (CI 1.53–3.00, *p* < 0.01). Additionally, among 315 patients, BRAF V600E-positive individuals had significantly higher FDG PET SUVs, with a mean difference of 5.1 (CI 4.3–5.8) compared to BRAF V600E-negative patients [37].

18F-FDG PET/CT helps in disease localization, recurrence detection, and therapy response assessment while providing additional information on tumor metabolism and aggressiveness [3,38,39]. This can be critical in patients with DTC with low thyroglobulin (Tg) levels. In the retrospective study by Lebbink and colleagues, FDG PET/CT was assessed for its effectiveness in detecting persistent or recurrent DTC in 27 patients who had low Tg levels (0.20–10.00 ng/mL) after undergoing thyroidectomy and I-131 ablation [38]. FDG PET/CT yielded positive results in 14 patients (51.9%), identifying lymph node metastases in 12 cases and lung metastases in 2 cases. DTC was confirmed in 13 of the 14 FDG PET/CT-positive cases. The test showed a sensitivity of 59.1%, a positive predictive value (PPV) of 92.9%, a specificity of 80.0%, and a negative predictive value (NPV) of 30.8% [38].

18F-FDG PET/CT can serve as an early predictor of response to multi-tyrosine kinase inhibitor (MKI) therapies, such as Lenvatinib, impacting overall survival in RAI-R DTC patients [40,41]. Valerio et al. demonstrated that [18F]-FDG PET/CT can serve as an early predictor of response to Lenvatinib treatment and is associated with overall survival outcomes in patients with RAI-R DTC [40]. In this study, 57.6% of patients (19 out of 33) showed the strongest metabolic response at the first [18F]-FDG-PET/CT scan conducted after 4 weeks of Lenvatinib treatment, while 15.1% of patients (5 out of 33) exhibited this response later. Additionally, 66.7% of patients demonstrated both an early metabolic response at the first [18F]-FDG-PET/CT scan and a morphological response at the first CT scan. Patients with a metabolic response on the final [18F]-FDG-PET/CT scan or a morphological response on the last CT scan had significantly longer median overall survival (OS): 40.00 vs. 8.98 months and 37.22 vs. 9.53 months, respectively [40]. Similarly, OS was longer in patients with a metabolic response at the 4-week [18F]-FDG-PET/CT scan, with survival of 36.53 vs. 11.28 months for those without this early response [40].

### 3.2. Ga-68-Based (Somatostatin Receptor) SSTR Imaging

#### 68Ga-DOTATATE/DOTANOC PET/CT

Gallium-68 (Ga-68)-labeled somatostatin analogs, such as Ga-68 DOTATOC and Ga-68 DOTATATE, enable the detection and localization of tumors expressing somatostatin receptors, aiding in diagnostic evaluations and treatment strategizing [42,43,44,45,46]. Ga-68 DOTATATE exhibits high affinity for (Somatostatin Receptor Subtype 2) SSTR2 [43,44,46]. While Ga-68 DOTATOC, Ga-68 DOTANOC, and Ga-68 DOTATATE can all engage with SSTR2, they demonstrate distinct affinity profiles towards other SSTR subtypes. Ga-68 DOTANOC shows significant affinity towards SSTR3 and SSTR5 [43,44,46].

These tracers have been compared to FDG-PET/CT. In a study by Li et al. involving 14 patients, 68Ga-DOTA-NOC PET/CT was positive in 12/14 cases (85.7%), while 18F-FDG PET/CT was positive in 13/14 cases (92.8%), with no significant difference (*p* = 0.204) [47]. Lesion-based analysis revealed that 18F-FDG PET-CT detected 80.7% (46/57) of lesions compared to 68Ga-DOTA-NOC PET-CT, which detected 63.2% (36/57) (*p* < 0.005) [47]. 18F-FDG PET-CT was superior for pulmonary metastasis, while 68Ga-DOTA-NOC PET-CT bettered in detecting bone metastasis, in that study.

Kreissl et al. observed that Ga-68-DOTATOC-PET detected new lesions unseen by FDG-PET; 69% of the patients with advanced RAI-refractory or -resistant FTC had SSTR expression [48]. In another study, a detection rate of 67% for metastatic lesions with Ga-68 DOTATATE PET/CT was observed that led to changes in management for 65.6% of patients (concordance between Ga-68 DOTATATE PET/CT and FDG-PET/CT was seen in 60/92 lesions; management alterations occurred in 21/32 patients for DOTATATE vs. 18/32 patients for FDG-PET) [49]. 68Ga-DOTATOC PET/CT was shown to be positive in 33% of the poorly differentiated/oxyphilic carcinoma cases in which FDG-PET/CT and RAI were negative [50].

These tracers show promise, although further studies are warranted.

### 3.3. 68Ga-DOTA-FAPI-04 PET/CT (Fibroblast Activation Protein Inhibitor PET/CT) and Similar Agents

Fibroblast activation protein (FAP), a serine protease, is selectively expressed in stromal fibroblasts linked to epithelial cancers but is low or absent in normal tissue fibroblasts [51]. A new class of molecules has been developed based on a quinoline-based FAP-specific inhibitor called UAMC1110 [52]. These molecules include FAPI-02, FAPI-04, FAPI-46, and other similar compounds [52,53,54]. They have been designed, synthesized, and tested in both preclinical and clinical settings. When labeled with gallium-68, these imaging probes show great promise because their biological half-life matches well with the physical half-life of gallium-68, making them effective for imaging purposes [52,53,54]. A novel FAP inhibitor, 68Ga-labeled DOTA.SA.FAPi, incorporating a squaric acid motif, has recently been developed [52,55].

68Ga-DOTA-FAPI-04 PET/CT and other similar agents target FAP that is expressed in cancer-associated fibroblasts, allowing for the visualization of cancerous and fibrotic tissues [56,57,58]. In some studies, 68Ga-DOTA-FAPI-04 PET/CT has been shown to be more sensitive and accurate in detecting metastatic lesions in thyroid cancer compared to 18F-FDG PET/CT, with a particular advantage in identifying metastases in lymph nodes and the lungs [54,57,59,60]. 68Ga-DOTA-FAPI-04 PET/CT exhibits high uptake values and provides better tumor-to-background contrast ratios [57,59].

68Ga-FAPI PET/CT has been shown to be effective in patients with recurrent PTC, often detecting more metastatic foci than 18F-FDG PET/CT [57]. 68Ga-FAPI PET/CT had a higher sensitivity than 18F-FDG PET/CT for depicting neck lesions (83% [65 of 78; 95% CI: 73, 90] vs. 65% [51 of 78; 95% CI: 54, 75], *p* = 0.01) and distant metastases (79% [87 of 110; 95% CI: 71, 86] vs. 59% [65 of 110; 95% CI: 50, 68], *p* < 0.001) [57].

In summary, FAP inhibitor-based imaging shows significant promise; however, further research is needed to validate its efficacy and clinical utility.

### 3.4. 18F-Sodium Fluoride PET/CT Imaging—Focus on Bony Metastatic Lesions

The utilization of 18F-sodium fluoride (NaF) PET/CT imaging in the detection of bony metastatic lesions, particularly in the context of advanced DTC, remains under investigation. Studies have indicated that bone PET/CT imaging may offer greater sensitivity and accuracy compared to conventional bone scintigraphy for detecting bone metastases in thyroid cancer [61]. Lee and colleagues compared the diagnostic accuracy of F-18 sodium fluoride PET/CT (bone PET/CT) with NM bone scintigraphy for detecting bone metastases in thyroid cancer patients. Among 17 suspected bone lesions, 10 were metastatic, and 7 were benign. Bone PET/CT demonstrated significantly higher sensitivity (100% vs. 20%, *p* = 0.008) and accuracy (82.4% vs. 41.2%, *p* < 0.025) than BS [61]. However, specificity was not significantly different between bone PET/CT (57.1%) and BS (71.4%, *p* > 0.05). Their study concludes that bone PET/CT is more sensitive and accurate than BS for detecting thyroid cancer bone metastases [61].

In patients with DTC, the utility of NaF PET/CT in detecting metastatic skeletal involvement compared to conventional post-therapy 131-I whole-body scans and 18F-FDG PET/CT remains inconclusive [62]. Some studies have reported that NaF PET/CT did not provide additional information regarding skeletal metastases compared to other imaging modalities [62].

### 3.5. PSMA-Based Imaging

#### 68Ga-PSMA and 18F-DCFPyL PET/CT

Prostate-specific membrane antigen (PSMA), also known as glutamate carboxypeptidase II (GCP II), is highly expressed in prostate cancer and has recently been found in the neovasculature of DTC [63]. PSMA imaging, utilizing radiotracers such as 68Ga-PSMA and 18F-DCFPyL, has shown utility for detecting advanced DTC, particularly in cases refractory to traditional therapies.

A study evaluated five patients with RAI-R DTC using 68Ga-PSMA PET/CT to determine their eligibility for 177Lu-PSMA-617 therapy (a Radiopharmaceutical agent targeting PSMA) [64]. All patients had suspicious tracer uptake for distant metastases, and three were eligible for therapy, with two patients receiving treatment [64]. One patient experienced disease progression within a month, with Tg levels increasing from 18 to 63 μg/L over the following months [64]. The other patient showed a partial but temporary response, with Tg levels initially decreasing from 17 to 9 μg/L but rising to 14 μg/L after 7 months as the disease progressed [64]. Another study showed that Ga-HBED-CC-PSMA PET/CT may identify patients who might qualify for PSMA-targeted radionuclide therapy [65]. In a study by our group, we showed that another PSMA imaging agent, (18)F-DCFPyl, may be valuable for identifying metastases in patients with metastatic RAI-refractory DTC by targeting and detecting neoangiogenesis within the tumor [66].

Despite the potential of PSMA-based imaging in DTC, the literature remains limited and heterogeneous [67]. Studies have reported variable detection rates of PSMA-targeted PET/CT in DTC patients, with some suggesting its use may not significantly impact patient management compared to FDG PET [67]. Prospective multicentric studies are needed to elucidate the true clinical utility of PSMA-based imaging in guiding treatment decisions and improving outcomes for patients with advanced DTC [68,69].

### 3.6. Arginine–Glycine–Aspartate (RGD)-Based Imaging

Integrins, composed of 18α and 8β subunits forming 24 transmembrane heterodimers, mediate cell adhesion and regulate key processes such as proliferation, migration, and survival, and among these, integrin αvβ3 is crucial for tumor neovascularization, growth, metastasis, and resistance to detachment-induced cell death [70]. Integrin-targeting probes based on the RGD motif exhibit high affinity for αvβ3 integrins, which are highly expressed in thyroid cancer, particularly on neovasculature and cancer cell surfaces, as shown by RNA sequencing data and thyroid cancer cell line studies [70,71]. RGD-based imaging in PET/CT utilizes radiolabeled peptides containing the RGD motif to target αvβ3 integrins [72,73].

#### 68Ga-DOTA-RGD2 PET/CT

Parihar et al. evaluated the diagnostic performance of 68Ga-DOTA-RGD(2) PET/CT as a neoangiogenesis imaging modality in 44 RAIR-DTC patients with negative post-therapy I-131 scans, comparing it with the standard (18)F-FDG PET/CT [74]. Sensitivity, specificity, and accuracy for 68Ga-DOTA-RGD(2) PET/CT were 82.3%, 100%, and 86.4%, respectively, while 18F-FDG PET/CT showed a similar sensitivity (82.3%) but lower specificity (50%) and accuracy (75%) [74]. The most common sites of disease were nodal metastases (82.9%) and thyroid bed lesions (10.5%). Among patients positive on 68Ga-DOTA-RGD(2) PET/CT, 82.1% exhibited radiotracer uptake exceeding liver levels [74]. A significant proportion of patients exhibiting positivity on 68Ga-DOTA-RGD2 PET/CT demonstrated high radiotracer uptake, suggesting the potential utility of novel 177Lu-based theranostics in this subgroup of patients that the authors demonstrated successfully on a patient [74,75]. While RGD-based tracers demonstrate superior tumor-to-background ratios in brain tumors compared to FDG, caution is warranted in interpreting liver lesions due to higher intensity in normal liver tissue [76].

Further details regarding the mechanisms of action and evidence supporting the indications for various radiopharmaceuticals in RAIR-DTC are summarized in Table 2.

## 4. Theranostics in Thyroid Cancer

Thyroid cancer management is gravitating towards personalized therapy guided by molecular imaging, with theranostics at its core [77]. Theranostics, a term used for combining diagnostic testing and targeted therapeutic interventions based on molecular targets, has been a part of thyroidology for over 70 years, particularly with radioiodine molecular imaging [78]. This approach allows for the tailoring of treatment strategies to individual patients, optimizing efficacy and minimizing adverse effects [78].

The foundational premise of radioactive iodine I-131 (RAI) therapy, dating back to 1946, remains pivotal in the multimodal management of DTC [79]. Leveraging the NIS expression in thyroid cells, RAI therapy targets residual and metastatic DTC cells, serving as a crucial determinant in the theranostic approach to DTC treatment [79].

Patients with metastatic DTC often present with negative diagnostic or post-therapy radioiodine scans, limiting the therapeutic use of 131-I. Redifferentiation plays a crucial role in advancing the treatment of radioiodine-refractory thyroid cancer (RAIR TC) by aiming to restore the tumor’s ability to take up and respond to radioiodine therapy. This strategy involves the use of specific drugs, such as MAPK pathway inhibitors, to reverse the dedifferentiation process that renders tumors resistant to standard radioiodine treatment. By reactivating key molecular pathways, redifferentiation can enhance radioiodine uptake in metastatic lesions, potentially improving the effectiveness of radioiodine therapy. This approach offers new hope for patients with RAIR TC, particularly those whose tumors regain sufficient radioiodine sensitivity, enabling the possibility of prolonged remission or even a cure. Redifferentiation not only expands therapeutic options but also helps refine the classification of RAIR TC by distinguishing between truly resistant cases and those responsive to this innovative treatment [80].

Tyrosine kinase inhibitors, commonly used in this setting, may fail due to resistance or intolerable side effects. A new approach, the redifferentiation strategy, aims to restore radioiodine sensitivity using specific drugs. This strategy has redefined RAIR TC to include two groups: persistent non-radioiodine-avid patients and “true” RAIR TC cases. The latter group shows restored radioiodine uptake in metastatic lesions but no radiological response, confirming true resistance to radioiodine therapy. Conversely, patients with restored uptake offer potential for remission or cure, similar to radioiodine-avid metastatic TC cases [81].

Redifferentiation therapies targeting specific genomic alterations may restore 131-I uptake, enabling further 131-I treatment [82]. In a review by Nostrand et al., 14 articles including prospective and retrospective studies that employed redifferentiation using seven agents across eight genetic alterations, with the largest studies involving 20–24 patients, were analyzed [82]. Among 101 treated patients, successful redifferentiation and response rates after 131-I therapy ranged from 33% to 100% [82].

New compounds like Lu-DOTAGA.(SA.FAPi)2 have shown potential as therapeutic options for aggressive refractory DTC cases, emphasizing the evolution of theranostics beyond conventional treatments [83]. In the study by Bilal et al., 15 patients (age: 55 ± 9 years) underwent treatment with [83] Lu-DOTAGA.(SA.FAPi)2, a radiotracer with a median whole-body effective half-life of 88.06 h [83]. A total of 45 treatment cycles were administered (median activity 8.2 ± 2.7 GBq) [83]. Tumor lesions had a median absorbed dose of 10.8 (IQR: 4.16–89.7) mSv/MBq per cycle. Serum Tg levels significantly decreased post-treatment (baseline median Tg: 10,549 ng/mL versus post-treatment: 5649 ng/mL). Molecular response showed partial response in four patients, and stable disease in three. Visual Analog Scale (VASmax) scores improved significantly as did the ECOG performance scores [83].

In summary, metastatic DTC, particularly when it progresses to a radioiodine-refractory state, presents challenges in management and prognosis [84]. Molecular imaging plays a pivotal role in evaluating and strategizing therapeutic approaches for such cases, offering insights into personalized therapeutic strategies grounded in molecular imaging [84].

Novel tracers like PSMA-based and RGD-based agents, alongside advancements in theranostics, present significant promise in improving the detection and characterization of DTC. These innovations provide unique insights into disease pathology, particularly in cases of radioiodine-refractory DTC, offering potential improvements in diagnostic precision and treatment outcomes. Additionally, integrating tracers such as [18F] TFB and 68Ga-DOTA-FAPI-04 could redefine the approach to managing metastases, facilitating personalized therapeutic strategies. Future applications should explore their integration into standard clinical practice, potentially enhancing the predictive and therapeutic landscape for DTC patients.

## 5. Future Directions—Application of Nanoparticles and Materials

While DTC generally carries a favorable prognosis, the process of dedifferentiation correlates with a less favorable clinical trajectory. Nanomaterials possess unique characteristics, such as multifunctionality, modifiability, and diverse detection capabilities, facilitating non-invasive and efficient diagnosis of thyroid cancer through multimodal imaging modalities [85]. Contrast-enhanced ultrasound imaging is highly accurate in diagnosing various tumors, particularly hepatic carcinoma, but is rarely applied to thyroid cancer. This study explored Src homology region 2 (SH2)-containing protein tyrosine phosphatase-2 (SHP)2-targeted PLGA nanoparticles encapsulating perfluoropentane for enhanced ultrasound molecular imaging [86]. SHP2 expression was significantly elevated in 65 thyroid tumor samples, as confirmed by immunohistochemistry, and these nanoparticles demonstrated specific tumor targeting and enhanced imaging signals in mouse models using LIFU (Low-intensity Focused Ultrasound) [86].

Another study compared methylene blue (and carbon nanoparticles as tracers for sentinel lymph node biopsy (SLNB) in 200 thyroid microcarcinoma patients [87]. The carbon nanoparticles demonstrated superior performance with higher sensitivity (93.3% vs. 80.6%), accuracy (97% vs. 93%), and a lower false-negative rate (5.2% vs. 9.9%) compared to methylene blue [87]. These findings suggest that carbon nanoparticles offer improved durability and accuracy for SLN imaging, better predicting cervical LN status and aiding in patient selection for dissection [87].

The future promise of nanotechnology in cancer imaging and therapy lies in its ability to integrate different functionalities into a single interventional approach, as demonstrated by the hydrogel-based nanotechnology described in this study [88]. By combining tumor localization, photon-to-heat conversion, and precise drug release, this approach enables multimodal imaging and localized therapeutic delivery [88].

## 6. Conclusions

In our review, we have summarized advancements in radiotracers and imaging technologies for the detection and management of advanced DTC. Traditional iodine-based radiopharmaceuticals, including I-123, I-131, and I-124 PET/CT, remain integral for imaging and therapy in radioiodine-avid DTC. These radiopharmaceuticals rely on the NIS for targeting thyroid cancer cells. I-124 PET/CT offers higher sensitivity and resolution compared to I-131 imaging, providing valuable information on metastases and aiding in optimizing therapeutic dosimetry. However, reduced or heterogeneous NIS expression in some DTC cases limits the effectiveness of these traditional agents.

18F-FDG PET/CT is widely used for the evaluation of radioiodine-refractory DTC, particularly in cases with aggressive tumor mutations like BRAF V600E. This imaging modality helps localize metabolically active lesions, detect recurrences, and assess therapy response. Gallium-68-based tracers such as Ga-68 DOTATATE and Ga-68 DOTATOC, targeting somatostatin receptors, are effective in identifying refractory and poorly differentiated DTC. Additionally, novel imaging agents like 68Ga-FAPI-04 PET/CT, targeting fibroblast activation protein in the tumor microenvironment, demonstrate high sensitivity and accuracy, particularly in detecting metastases in lymph nodes and lungs.

Theranostic approaches integrating diagnostic imaging and targeted therapies are advancing the management of DTC. PSMA-based imaging has shown utility in identifying neovascularization, while RGD-based imaging targets integrin expression associated with tumor progression. Nanotechnology offers potential for further innovation through multimodal imaging and localized therapeutic delivery. These advancements contribute to more precise diagnostics and tailored treatment strategies for advanced and refractory DTC.

## Figures and Tables

**Table 2 cancers-17-00425-t002:** Mechanism of action and evidence of indication of various radiopharmaceuticals in RAIR-DTC.

Radiopharmaceuticals for RAIR-DTC			
Radiotracer	Mechanism of Action	Evidence	Indication
18F-FDG PET/CT	Targets glucose metabolism; identifies metabolically active tumors, especially in dedifferentiated and aggressive forms of DTC.	Helps in disease localization, recurrence detection, and therapy response assessment in RAI-refractory DTC.	Used to detect metastases and monitor therapy in radioiodine-refractory DTC.
68Ga-DOTATATE/DOTANOC PET/CT	Binds to somatostatin receptors (SSTR2 and SSTR3); highlights neuroendocrine differentiation.	Demonstrated higher sensitivity for metastatic lesions and better performance in detecting bone metastases compared to FDG-PET.	Used for imaging neuroendocrine differentiation in poorly differentiated or iodine-refractory thyroid cancers.
68Ga-DOTA-FAPI-04 PET/CT	Targets fibroblast activation protein in the tumor microenvironment.	Provides better tumor-to-background contrast than FDG-PET for detecting metastatic lesions, particularly in the lymph nodes and lungs.	Useful for detecting recurrent DTC and metastatic lesions, especially in areas missed by other imaging modalities.
18F-Sodium Fluoride PET/CT	Highlights increased bone turnover in metastatic sites by binding to hydroxyapatite in bone lesions.	More sensitive and accurate than bone scintigraphy for detecting bony metastases in thyroid cancer.	Used for identifying skeletal metastases in advanced DTC.
68Ga-PSMA and 18F-DCFPyL PET/CT	Targets prostate-specific membrane antigen (PSMA) expressed in tumor neovasculature.	Identifies neovascularization in metastatic lesions, including RAI-refractory cases, and may be useful in guiding PSMA-targeted radionuclide therapy.	Used to guide treatment in metastatic RAI-refractory DTC patients.
68Ga-DOTA-RGD2 PET/CT	Binds to integrin αvβ3, critical for tumor angiogenesis and metastasis.	Demonstrated high sensitivity and specificity for identifying RAI-refractory lesion.	Useful for neoangiogenesis imaging and potential theranostic applications in advanced DTC.

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
