# Peer review of "Current and Emerging Radiotracers and Technologies for Detection of Advanced Differentiated Thyroid Cancer: A Narrative Review"

_cancers, 2025, doi:10.3390/cancers17030425_

Round 1
Reviewer 1 Report
Comments and Suggestions for Authors
Dear Authors,
this paper focuses on an interesting and open field of research. Different issues are present:
- the title is somehow misleading since half of the paper focuses on well established tracers used in DTC and not on "emerging" ones;
- the abstract is missing;
- it is mandatory to specify that this is a narrative review;
- different abbreviations have been used without specifying their meaning (for example RNA, BRAF, ..);
- some parts need some syntactic revision;
- the paragraph "NIS (Sodium-Iodide Symporter)-Based Imaging: Iodine-Based Radiotracers (I-123, I-131, I-124 29 PET/CT)" has some major problems and in general is hard to understand the whole meaning. For example, it is not specified that it focuses on planar imaging and not SPECT/CT, it is not clear why the guidelines have been cited and their impact in this setting, etc.. In general, the cited papers seem not to clearly summarize the actual knowledge in this field (this consideration could however apply to the whole text);
- Table 1 has only references for the 123I part and not for the section regarding 131I;
- in the "18. F-FDG PET/CT" section the role of this imaging modality in Tg negative patients is presented. However, current guidelines suggest to use this modality in patients with negative iodine imaging but positive Tg levels. It is necessary to discuss or clarify this fact;
- in general, the sections "Theranostics in Thyroid Cancer" and "Future Directions-Application of Nanoparticles and materials" are the most interesting of the paper and best fit with its title. Maybe it could be useful to consider to focus on these two parts.
- a general discussion of the data proposed in the review is missing. In particular, this section should focus on the value of new tracers for the assessment on DTC, their actual role and on their future applications.
Author Response
1- the title is somehow misleading since half of the paper focuses on well-established tracers used in DTC and not on "emerging" ones; We changed the title to: Current and Emerging Radiotracers and technologies for Detection of Advanced Differentiated Thyroid Cancer: A Narrative Review
2- it is mandatory to specify that this is a narrative review; We mentioned it at the end of the title
3- different abbreviations have been used without specifying their meaning (for example RNA, BRAF,.); Some of the more uncommon abbreviations were specified in the draft.
4- some parts need some syntactic revision; We revised the article.
5- the paragraph "NIS (Sodium-Iodide Symporter)-Based Imaging: Iodine-Based Radiotracers (I-123, I-131, I-124 29 PET/CT)" has some major problems and in general is hard to understand the whole meaning. For example, it is not specified that it focuses on planar imaging and not SPECT/CT, it is not clear why the guidelines have been cited and their impact in this setting, etc. In general, the cited papers seem not to clearly summarize the actual knowledge in this field (this consideration could however apply to the whole text); Some part of this paragraph was revised to make it easier to understand.
6- Table 1 has only references for the 123I part and not for the section regarding 131I; The references mentioned in I-123 part are applied to both I-123 and I-131 from the same papers.
7- in the "18. F-FDG PET/CT" section the role of this imaging modality in Tg negative patients is presented. However, current guidelines suggest using this modality in patients with negative iodine imaging but positive Tg levels. It is necessary to discuss or clarify this fact; We discussed the role of 18-F-FDG PET/CT in the study conducted by Lebbink and colleagues in patients with low Tg levels. But as the respected reviewer mentioned, current guidelines suggest being used in Tg positive patients.
8- in general, the sections "Theranostics in Thyroid Cancer" and "Future Directions-Application of Nanoparticles and materials" are the most interesting of the paper and best fit with its title. Maybe it could be useful to consider focusing on these two parts. We agree with the respected reviewer.
9- A general discussion of the data proposed in the review is missing. In particular, this section should focus on the value of new tracers for the assessment on DTC, their actual role and on their future applications. The following paragraph was added to the article:
Novel tracers like PSMA-based and RGD-based agents, alongside advancements in theranostics, present significant promise in improving the detection and characterization of DTC. These innovations provide unique insights into disease pathology, particularly in cases of radioiodine-refractory DTC, offering potential improvements in diagnostic precision and treatment outcomes. Additionally, integrating tracers such as [18F] TFB and 68Ga-DOTA-FAPI-04 could redefine the approach to managing metastases, facilitating personalized therapeutic strategies. Future applications should explore their integration into standard clinical practice, potentially enhancing the predictive and therapeutic landscape for DTC patients.
Reviewer 2 Report
Comments and Suggestions for Authors
I thought this is a solid, well-written review. The reviewer appreciates the effort that went into to updating the literature on this topic.
minor comments and suggestions:
1) Not every reader for this is going to have enough baseline information on the topic of differentiated thyroid cancer. I would suggest that the authors add in either into the introduction and/or as supplementary materials for following information
a) etiology, pathophysiology, , prognosis, risk factors, typical treatment path, and population information about differentiated thyroid cancer even biological sex.
b) It may be also informative to discuss how its 'path' is different than other thyroid cancers, including MTC, ATC< and Hurthle cell
2) It may be nice to separate and identify which of the radiopharmaceuticals are 'diagnostic in nature' vs treatment. How are they used in other parts of the paper (if at all?)
3) Multikinases are mentioned, in the Radioiodine-Refractory DTC
4) What about anti-angiogenic multikinase therapies that are emerging? Is there a role?
5) more on the redifferentiation pathophysiology would be helpful (either in supplementary as notes, and/or in the article)
6) It would help to have more tables to go along with your dense narrative information. It should be pretty easy to generate those tables for each section, it would add readability and reach of the article (not everyone will be able to grab all the important ideas you are sharing.
The authors have conducted extensive work and its quality would be very helpful for readership as its an update all in one spot. Appreciate the quality and content. Perhaps, the authors should also consider sharing continuing to share their work in oncology-focused conferences such as American society of clinical oncology (ASCO) and perhaps partner with more oncologists, pathologists and radiologists and/or present in the American Thyroid Association, the Mazzaferri Thyroid Cancer Conference and/or the UPMC Conference so their work could receive more dissemination and feedback.
Author Response
1) Not every reader about this is going to have enough baseline information on the topic of differentiated thyroid cancer. I would suggest that the authors add in either into the introduction and/or as supplementary materials for the following information
- a) etiology, pathophysiology, prognosis, risk factors, typical treatment path, and population information about differentiated thyroid cancer even biological sex: Thank you for this suggestion. The authors' decision to keep the article concise and focused on the radiology and imaging aspects of advanced DTC ensures clarity and precision in addressing the primary topic. This streamlined approach enhances the article's relevance to its intended audience.
- b) It may be also informative to discuss how its 'path' is different than other thyroid cancers, including MTC, ATC< and Hurthle cell: The following information was added to the article: DTC, which includes papillary and follicular thyroid cancers, differs significantly in pathology compared to other thyroid malignancies such as MTC, ATC, and Hurthle cell carcinoma. DTC arises from thyroid follicular cells and is characterized by its well-differentiated cellular architecture and a generally indolent course. It typically retains the ability to uptake iodine due to the presence of sodium-iodide symporters, making radioiodine therapy a viable treatment. In contrast, MTC originates from parafollicular C-cells and is associated with calcitonin secretion, lacking the iodine uptake pathway. ATC represents an undifferentiated, highly aggressive form of thyroid cancer with rapid growth, resistance to most therapies, and extremely poor prognosis. Hurthle cell carcinoma, a variant of follicular thyroid cancer, is noted for its abundant mitochondria and eosinophilic cytoplasm, often displaying a more aggressive clinical course than conventional DTC. These distinct pathological features among thyroid cancer subtypes underpin differences in clinical presentation, diagnostic approaches, and treatment strategies.
2) It may be nice to separate and identify which of the radiopharmaceuticals are 'diagnostic in nature' vs treatment. How are they used in other parts of the paper (if at all?) The following information was incorporated into the manuscript: Diagnostic radiopharmaceuticals include I-123, I-124 PET/CT, and novel tracers like [18F] TFB, [18F] FSO3, and 68Ga-based somatostatin receptor (SSTR) imaging agents, which are primarily utilized to visualize disease burden, assess metastases, and guide therapeutic decisions. On the other hand, therapeutic radiopharmaceuticals, such as I-131 and emerging agents like 177Lu-DOTA-RGD2 and 68Ga-DOTA-FAPI-04, are designed to deliver targeted radiation to cancer cells, capitalizing on molecular and cellular pathways unique to thyroid cancer.
3) Multikinases are mentioned, in the Radioiodine-Refractory DTC
We have removed the references and the paragraphs. We agree it may not sync with the thrust of the manuscript.
4) What about anti-angiogenic multikinase therapies that are emerging? Is there a role? Yes, emerging anti-angiogenic MKIs are being investigated for the treatment of RAIR DTC.
- Anlotinib: A recent phase II randomized, double-blind, placebo-controlled trial demonstrated that anlotinib significantly prolonged progression-free survival in patients with locally advanced or metastatic RAIR DTC, indicating its potential as a therapeutic option.
- Apatinib: Primarily used for advanced gastric cancer, apatinib has shown promise in RAIR DTC. Studies suggest it may improve progression-free survival, though further research is needed to confirm its efficacy and safety in this specific context.
These emerging MKIs offer potential new avenues for managing RAIR DTC, especially for patients who have developed resistance to existing therapies.
5) more on the redifferentiation pathophysiology would be helpful (either in supplementary as notes, and/or in the article): 2 following paragraphs were added to the article.
Managing RAIR TC is complex. Redifferentiation plays a crucial role in advancing the treatment of RAIR TC by aiming to restore the tumor's ability to take up and respond to radioiodine therapy. This strategy involves the use of specific drugs, such as MAPK pathway inhibitors, to reverse the dedifferentiation process that renders tumors resistant to standard radioiodine treatment. By reactivating key molecular pathways, redifferentiation can enhance radioiodine uptake in metastatic lesions, potentially improving the effectiveness of radioiodine therapy. This approach offers new hope for patients with RAIR TC, particularly those whose tumors regain sufficient radioiodine sensitivity, enabling the possibility of prolonged remission or even a cure. Redifferentiation not only expands therapeutic options but also helps refine the classification of RAIR TC by distinguishing between truly resistant cases and those responsive to this innovative treatment.
Tyrosine kinase inhibitors, commonly used in this setting, may fail due to resistance or intolerable side effects. A new approach, the redifferentiation strategy, aims to restore radioiodine sensitivity using specific drugs. This strategy has redefined RAIR TC to include two groups: persistent non-radioiodine-avid patients and "true" RAIR TC cases. The latter group shows restored radioiodine uptake in metastatic lesions but no radiological response, confirming true resistance to radioiodine therapy. Conversely, patients with restored uptake offer potential for remission or cure, similar to radioiodine-avid metastatic TC cases.
6) It would help to have more tables to go along with your dense narrative information. It should be pretty easy to generate those tables for each section, it would add readability and reach of the article (not everyone will be able to grab all the important ideas you are sharing: A table was created to summarize the information about RAI-R DTC.
The authors have conducted extensive work, and its quality would be very helpful for readership as it’s an update all in one spot. I appreciate the quality and content. Perhaps, the authors should also consider sharing continuing to share their work in oncology-focused conferences such as American society of clinical oncology (ASCO) and perhaps partner with more oncologists, pathologists and radiologists and/or present in the American Thyroid Association, the Mazzaferri Thyroid Cancer Conference and/or the UPMC Conference so their work could receive more dissemination and feedback.
Round 2
Reviewer 1 Report
Comments and Suggestions for Authors
Thank you for consider the suggestions and improve the quality of the paper.